

# Variation and interrelationships in the growth, yield, and lodging of oat under different planting densities

Lingling Liu, Guoling Liang, Wenhui Liu and Zeliang Ju

[1] College of Animal Husbandry and Veterinary Sciences, Qinghai University, Key Laboratory for Utilization of Superior Forage Germplasm Resources in the Qinghai-Tibet Plateau, Qinghai Province, Xining, Qinghai, China

[2] Laboratory for Research and Utilization of Qinghai Tibet Plateau Germplasm Resources, Xining, Qinghai, China

Corresponding author
Guoling Liang,
qhliangguoling@163.com

## ABSTRACT

**Background**. Oat is a dual-purpose cereal used for grain and forage. The demand of oat has been increasing as the understanding of the nutritional, ecological, and economic values of oat increased. However, the frequent lodging during the growing period severely affect the high yielding potential and the quality of the grain and forage of oat.

**Methods**. Therefore, we used the lodging-resistant variety LENA and the lodging-sensitive variety QY2 as materials, implementing four different planting densities: $2.25 \times 10^6$ plants/ha (D1), $4.5 \times 10^6$ plants/ha (D2), $6.75 \times 10^6$ plants/ha (D3), and $9 \times 10^6$ plants/ha (D4). At the appropriate growth and development stages, we assessed agronomic traits, mechanical characteristics, biochemical compositions, yield and its components. The study investigated the impact of planting density on the growth, lodging, and yield of oat, as well as their interrelationships. Additionally, we identified the optimal planting density to establish a robust crop structure. The research aims to contribute to the high-yield and high-quality cultivation of oat.

**Results**. We observed that with increasing planting density, plant height, grass and grain yields of both varieties first increased and then decreased; root fresh weight, stem diameter, stem wall thickness, stem puncture strength, breaking strength, compressive strength, lignin and crude fiber contents, and yield components decreased; whereas the lodging rate and lodging coefficient increased. Planting density affects lodging by regulating plant height, height of center of gravity, stem wall thickness, internode length, and root fresh weight of oat. Additionally, it can impact stem mechanical strength by modulating the synthesis of lignin and crude fiber, which in turn affecting lodging resistance. Plant height, height of center of gravity, stem wall thickness, internode length, root fresh weight, breaking strength, compressive strength, lignin and crude fiber content, single-plant weight, grain yield and 1,000-grain weight can serve as important indicators for evaluating oat stem lodging resistance. We also noted that planting density affected grain yield both directly and indirectly (by affecting lodging); high density increased lodging rate and decreased grain yield, mainly by reducing 1,000-grain weight. Nonetheless, there was no significant relationship between lodging and grass yield. As appropriate planting density can increase the yield while maintaining good lodging resistance, in this study, $4.5 \times 10^6$ plants/ha (D2) was found to be the best planting density for oat in terms of lodging resistance and grass and grain yield. These findings can be used as a reference for oat planting.

# INTRODUCTION

Oat (*Avena sativa* L.) is an important source of forage that contributes to the sustainable development of the livestock industry. Furthermore, the grains of oat are rich in nutrients, and people are increasingly interested in them as health food (*Muhammad et al., 2020*; *Ma et al., 2017*). According to statistics, the global oat cultivation area reached 9.492 million hectares in 2021, with China accounting for 5.53%. And the cultivation area continues to expand, indicating a promising outlook for the oat industry (https://www.huaon.com/channel/trend/892251.html). However, in the process of oat production, extreme climatic conditions and poor stress resistance of oat varieties lead to frequent lodging, which seriously limits oat yield, quality, and mechanized harvesting. In addition, due to the limitation of planting technology, irrigation facilities, fertilizers, and the mechanization process, oat cultivation depends largely on the increase in planting density, but high planting density increase the risk of lodging and eventually reduce the economic efficiency of oat production (*Liu et al., 2021*).

Lodging occurs after the appearance of the panicle and is more common during the grain filling and maturing stages (*Wang et al., 2021*). It can damage plant spatial distribution and population structure, reduce photosynthetic performance of leaves, deteriorate the local microenvironment, increase diseases, decrease harvest index, increase production costs, and impair yield and quality (*Wu & Ma, 2019*; *Zhang et al., 2020*). The mechanism underlying lodging is extremely complex, and its determinants fall into three categories-environment, cultivation measures, and genetic constitution (*Liu et al., 2022*; *Wu et al., 2022*). While humans are largely unable to control the influence of environmental factors (wind, rain, and temperature), there are several other challenges involved in breeding varieties to change the genetic characteristics of the crop. Therefore, improving production performance and lodging resistance through modified cultivation measures is the current focus of oat production and an effective way to achieve high-yield and high-quality for oat.

Planting density plays an important role in improving crop yield and coordinating crop growth. If planting density is too low, plants cannot effectively use land and environmental resources, which leads to wastage of resources and low crop yield. In contrast, high planting density affects crop population structure and nutrient uptake, resulting in weak stems and a reduced root system, increasing the risk of lodging and ultimately affecting yield and quality (*Ahmad et al., 2021*; *Khan et al., 2018*). Plant height, height of center of gravity, root size, and diameter, wall thickness, mechanical strength, lignin and cellulose contents of stems have been shown to influence crop lodging resistance, where planting density influences lodging by regulating these morphological or physiological characteristics. Reportedly, compared to lodging-sensitive varieties, lodging-resistant varieties exhibit lower plant height, height of center of gravity, and increased diameter, wall thickness, mechanical strength, and lignin and cellulose content of stems (*Tian et al., 2015*; *Silveira et al., 2022*).

Many studies have been conducted to optimize crop growth, lodging and yield by regulating planting density in various crops, such as maize (*Jia et al., 2018*), wheat (*Luo et al., 2022*), and rapeseed (*Kuai et al., 2016*). However, the related researches on oat are still limited, which largely hinders its economic potential. Therefore, in this study, we selected two oat varieties with different lodging resistance and combined them with various planting densities. We investigated the variations in growth, yield and lodging of oat under different planting densities, and explored the interrelationships among them. Furthermore, we determined the optimum planting density for oat production. The findings of this study aim to improve the growth performance of oat varieties in an adequate planting density, to achieve high quality and yield and provide a practical basis for increased oat production.

## MATERIALS & METHODS

### Experimental site

The experiment was conducted in Xining City, Qinghai Province, China (101°33′20″E, 36°30′57″N), with an average altitude of 2,592 m above sea level and a plateau continental-type climate characterized by a cold and humid, but no absolute frost-free period. The average annual temperature was 5.1 °C, the average annual precipitation was 510 mm (mostly concentrated in July–September), and the average annual evaporation was 1,830 mm.

### Experimental design and field management

The two oat varieties used in this study—lodging-resistant variety LENA and lodging-sensitive variety QY2—were obtained from the Qinghai Academy of Animal and Veterinary Science. A two-year (2018–2019) field experiment was conducted using a randomized block design, and the four planting densities used for this experiment were $2.25 \times 10^6$ plants/ha (D1), $4.5 \times 10^6$ plants/ha (D2), $6.75 \times 10^6$ plants/ha (D3), and $9 \times 10^6$ plants/ha (D4). The actual sowing quantity of each variety was calculated based on germination rate and 1,000-grain weight. For LENA, the average 1,000-grain weight was 27.77 g, the germination rate was 95%, and its sowing quantities were 65.7 kg/ha (D1), 131.6 kg/ha (D2), 197.3 kg/ha (D3), and 263.1 kg/ha (D4). In contrast, for QY2, the average 1,000-grain weight was 32.20 g, the germination rate was 95%, and its sowing quantities were 76.2 kg/ha (D1), 152.6 kg/ha (D2), 228.8 kg/ha (D3), and 305.1 kg/ha (D4). The area of each plot was 15 m² (3 m × 5 m; $n = 3$), with row spacing of 20 cm and block spacing of 1 m. Furthermore, 150 kg/ha diammonium phosphate and 75 kg/ha urea were applied as base fertilizers before sowing, and plots were hand-weeded at the tillering stage. The previous crop cultivated on the plot was oilseed rape (*Brassica napus* L.).

### Plant sampling and measurements
#### Main agronomic traits
At the milk stage, 12 uniform plants were randomly selected from each planting-density plot to measure the following agronomic traits.

Plant height (PH): Distance from the base of the plant to the highest point at the top.

Height of center of gravity (HCG): Distance from the base of the stem (with leaves, sheaths, and spikes) to the equilibrium pivot point after the stem was balanced by placing the main stem at a pivot point.

Root fresh weight (RFW): The fresh weight of the underground portion of the plant. Excavated the complete plants, rinsed the roots with clean water, dried the surface moisture with absorbent papers, and then weighed using an LED-series electronic scale (HZ Corporation Co., Ltd.).

Above-ground fresh weight (AFW): The fresh weight of the above-ground portion of the plant, which consists of the stems, leaves, sheaths, and spikes.

Length of the second (third) stem internode (SL2/SL3): The length from the first (second) stem node to the second (third) stem node.

Diameter of the second (third) stem internode (SD2/SD3): Diameter at the middle of the second (third) stem internode.

Wall thickness of the second (third) stem internode (WT2/WT3): We cut the oat stems at the middle of the second (third) stem internodes and subsequently measured the wall thickness using a vernier caliper (Shanghai SANTO Technology Co. Ltd., Shanghai, China).

### Mechanical characteristics

At the milk stage, 12 uniform plants were randomly selected from each planting-density plot, and a YYD-1 strength tester (Zhejiang Top Technology Co. Ltd., Hangzhou, China) was used to measure the puncture strength, breaking strength, and compressive strength of the second and third stem internodes.

Puncture strength (PS): A puncture probe with a cross-sectional area of one mm$^2$ was used for measuring puncture strength. The stems (without leaf sheaths) were placed in the groove of the tester, with a distance of two cm between the two points, and inserted vertically downward into the middle of the internode at a constant speed. The maximum force required by the probe to puncture the epidermis of the stem was recorded as the puncture strength.

Breaking strength (BS): A bending probe was used to determine breaking strength. The protocol was the same as that for the determination of puncture strength, *i.e.,* the maximum force that broke the stem was recorded as the breaking strength.

Compressive strength (CS): A compressive probe was used to estimate compressive strength. The protocol was the same as that for the determination of puncture strength, *i.e.,* the maximum force that bent the stem was recorded as the compressive strength.

### Lodging

The field lodging rate (FLR) was determined at the maturity stage, and the lodging index (LI) and lodging coefficient (LC) were calculated as follows:

$$FLR(\%) = (\text{lodging area})/(\text{plot area}) * 100 \ (\textit{Peltonen-Sainio \& Jarvinen, 1995}). \quad (1)$$

$$LI = (HCG * AFW)/BS2 \ (\textit{Wang et al., 2015}). \quad (2)$$

$$LC = (PH * AFW)/(RFW * BS2) \ (\textit{Wang \& Du, 2001}). \quad (3)$$

where BS2, breaking strength of the second stem internode.

### Biochemical compositions

Oat plants at the milk stage were selected, and the second and third stem internodes above the ground were first oven-dried at 105 °C for 30 min and then at 65 °C to constant weight (*Argenta et al., 2021*). Thereafter, they were ground and passed through a 60-mesh sieve for the determination of crude fiber and lignin contents.

Crude fiber content (CF) determination: The method used in *Zakirullah et al. (2017)* was modified and applied. We took approximately 1.0 g of the sample (W1) in a 250-mL beaker and added 1.25% $H_2SO_4$ to make the volume up to 200 mL. The mixture was digested by micro-boiling (95 °C) for 30 min, and then filtered and washed. Subsequently, we added 1.25% NaOH and made up the volume up to 200 mL. Then, we heated (98 °C) the mixture for 30 min and filtered and washed the residue. This residue was placed in a pre-weighed crucible and then in an oven at 105 °C for 24 h for drying. After recording the dry weight (W2), the sample was placed in a muffle furnace at 600 °C for 4 h and weighed after cooling (W3). Finally, the following formula was used to calculate the crude fiber content:

$$\text{Crude fiber } (\%) = (W2 - W3)/W1 * 100. \tag{4}$$

Lignin content (LN) determination: The method used in *Brinkmann, Blaschke & Polle (2002)* was modified and applied. We used approximately 0.5 g of the sample (W1) in a 250-mL beaker, added 100 mL of 0.5 M $H_2SO_4$ (containing 1 g of cetyltrimethylammonium bromide), and boiled the mixture for 1 h under continuous stirring. A drop of octan-2-ol was added as an antifoam agent. We filtered and washed the mixture 3-5 times with distilled water and then washed it with acetone until further decoloration was not observed. The residue was dried at 105 °C for 2 h, followed by the addition and mixing of 10 mL of 72% $H_2SO_4$ and then another 10 mL of 72% $H_2SO_4$ after 1 h for continued hydrolysis for 3 h. The residue was then washed with distilled water until it was acid-free, dried at 105 °C for 2 h, cooled, and weighed (W2). The residue was placed in a muffle furnace at 500 °C for 3 h, cooled, and weighed again to determine ash content (W3). Lignin content was then calculated as follows:

$$\text{Lignin } (\%) = (W2 - W3)/W1 * 100. \tag{5}$$

## Yield and yield component measurements

Single-plant weight, fresh grass yield, and hay yield were determined at the milk stage. 15 plants were randomly selected from each planting-density plot to measure single-plant weight, after removing 40 cm of the boundary per plot. Half of the plants in the plots were harvested and tied into bundles, and these bundles were weighed separately with an electronic balance to determine the fresh grass yield. About 1 kg of the fresh grass samples were taken from each plot, oven-dried first at 105 °C for 30 min and then at 65 °C to constant weight, to determine hay yield. The results were converted into tons per hectare. The remaining half of the plants was harvested at maturity to determine grain yield and

its components. We also randomly selected 15 plants from each planting-density plot to measure the length of the main panicle (distance from the base to the top of the main panicle), number of grains per plant, weight of grains per plant, and 1,000-grain weight. Grains were harvested, dried in natural light, and weighed to convert yields in tons per hectare.

## Data analyses

Data were analyzed using Microsoft Excel 2010 and SPSS Statistics 22.0. Duncan's test (at $P < 0.05$) was applied to compare the significance of characteristic means, and analysis of variance was performed using the general linear model. Additionally, SPSSPRO was used for Pearson's correlation and regression analyses, and OriginPro 2021 (OriginLab Corporation, Northampton, MA, USA) was used for generating graphs.

## RESULTS

### Effects of planting density on agronomic traits

The diameter, wall thickness and length of stem internode, plant height, height of center of gravity and root fresh weight of the two oat varieties at different planting densities are shown in Table 1. Generally, compared to the lodging-sensitive variety QY2, the lodging-resistant variety LENA exhibited lower values of plant height, length of the second and third stem internodes, height of center of gravity, but higher values of root fresh weight, diameter and wall thickness of the second and third stem internodes in two years. With increasing planting density, the plant height of both LENA and QY2 first showed an increasing trend and then a decreasing trend, and reached maximum plant height at D2, whereas diameter and wall thickness of the second and third stem internodes, and root fresh weight showed a decreasing trend. The trends in length of the second and third stem internodes, and height of center of gravity were not consistent between the two varieties.

### Effects of planting density on mechanical characteristics of the stem

Analysis of the mechanical characteristics of the stems of both varieties revealed that the puncture strength, breaking strength, and compressive strength of the second and third stem internodes showed a decreasing trend with increasing planting density in two years (Fig. 1). LENA exhibited higher puncture strength, breaking strength, and compressive strength of the second and third stem internodes compared to QY2. Moreover, at the same planting density, the puncture strength, breaking strength, and compressive strength of the second stem internode were higher than those of the third stem internode for both varieties, indicating that the second stem internode had stronger mechanical strength.

### Effects of planting density on biochemical compositions

The lodging-resistant variety LENA exhibited higher lignin and crude fiber contents than the lodging-sensitive variety QY2 in two years (Figs. 2A and 2B). With increasing planting density, the lignin and crude fiber contents of the second and third stem internodes of both varieties tended to decrease. Nonetheless, at the same planting density, both varieties exhibited higher lignin and crude fiber contents in the second stem internode compared to the third stem internode.

Table 1 Effects of planting density on agronomic traits of the two oat varieties grown in 2018 and 2019.

| Years | Varieties | D | PH | SD2 | SD3 | WT2 | WT3 | SL2 | SL3 | HCG | RFW |
|---|---|---|---|---|---|---|---|---|---|---|---|
| 2018 | LENA | D1 | 1.01a | 4.26a | 4.94a | 1.08a | 0.85a | 7.02b | 13.65a | 0.36ab | 1.81a |
| | | D2 | 1.05a | 3.92b | 4.44ab | 0.95b | 0.75b | 7.32b | 14.18a | 0.38a | 1.48b |
| | | D3 | 1.01a | 3.56c | 3.94b | 0.89b | 0.71bc | 8.04ab | 14.22a | 0.36ab | 1.28b |
| | | D4 | 0.94b | 2.82d | 3.07c | 0.77c | 0.67c | 8.61a | 15.21a | 0.33b | 0.71c |
| | QY2 | D1 | 1.28ab | 4.33a | 4.42a | 0.58a | 0.46a | 21.35a | 30.95a | 0.51ab | 1.86a |
| | | D2 | 1.29a | 3.75b | 4.01b | 0.55a | 0.46a | 16.13c | 23.95b | 0.52a | 1.20b |
| | | D3 | 1.25ab | 3.49c | 3.77b | 0.46b | 0.45a | 18.86b | 23.50b | 0.50ab | 0.92c |
| | | D4 | 1.22b | 3.45c | 3.66b | 0.44b | 0.44a | 18.23b | 24.16b | 0.49b | 0.75d |
| 2019 | LENA | D1 | 1.25a | 4.41a | 5.05a | 1.14a | 0.90a | 7.53a | 13.86b | 0.48a | 1.98a |
| | | D2 | 1.28a | 3.99ab | 4.51b | 1.07b | 0.81b | 7.67a | 14.48b | 0.50a | 1.66b |
| | | D3 | 1.23a | 3.67b | 4.01b | 0.97c | 0.79c | 8.37a | 14.56b | 0.47a | 1.43c |
| | | D4 | 1.12b | 2.92c | 3.29c | 0.92c | 0.71d | 9.16a | 15.34a | 0.47a | 0.77d |
| | QY2 | D1 | 1.36a | 3.84a | 4.25a | 0.76a | 0.65a | 12.25a | 26.50a | 0.58a | 0.92a |
| | | D2 | 1.38a | 3.33b | 3.61b | 0.74a | 0.63ab | 12.44a | 22.00b | 0.57a | 0.64b |
| | | D3 | 1.34a | 3.21bc | 3.58bc | 0.68b | 0.62ab | 11.66a | 22.21b | 0.57a | 0.55c |
| | | D4 | 1.29b | 2.85c | 3.30c | 0.62c | 0.57b | 8.01b | 22.13b | 0.54b | 0.30d |

Notes.
D, density; PH, plant height (m); SD2 and SD3, diameter of the second and third stem internodes (mm); WT2 and WT3, wall thickness of the second and third stem internodes (mm); SL2 and SL3, length of the second and third stem internodes (cm); HCG, height of center of gravity (m); RFW, root fresh weight (g).
Different letters represent significant differences at $P < 0.05$.

## Effects of planting density on lodging

We observed that the lodging-resistant variety LENA exhibited much lower field lodging rate and lodging coefficient than the lodging-sensitive variety QY2 in two years (Fig. 3). The field lodging rate of both varieties showed an increasing trend with increasing planting density, but their differences at planting densities D1 and D2 were not significant (Fig. 3A). The tendency of change in lodging index of the two varieties was different; LENA first showed an increasing trend and then a decreasing trend, reaching its maxima in D2, whereas QY2 reached its maxima in D4, and with no significant differences between the planting densities (Fig. 3B). Additionally, the lodging coefficient of both varieties increased with increasing planting density (Fig. 3C).

## Effects of planting density on yield and its components
### Grass yield and its components

Analyses of single-plant weight, fresh grass yield, and hay yield of the two oat varieties revealed a decreasing trend in the single-plant weight of both varieties, whereas fresh grass yield and hay yield first exhibited an increasing trend and then a decreasing trend with increasing planting density in two years (Table 2). In 2018, the fresh grass yield and hay yield of LENA reached maxima in D4, while QY2 reached maxima in D3, both of them had minima in D1. In 2019, the fresh grass yield and hay yield of the two varieties reached maxima in D3 and minima in D1, and the difference was significance ($P < 0.05$).

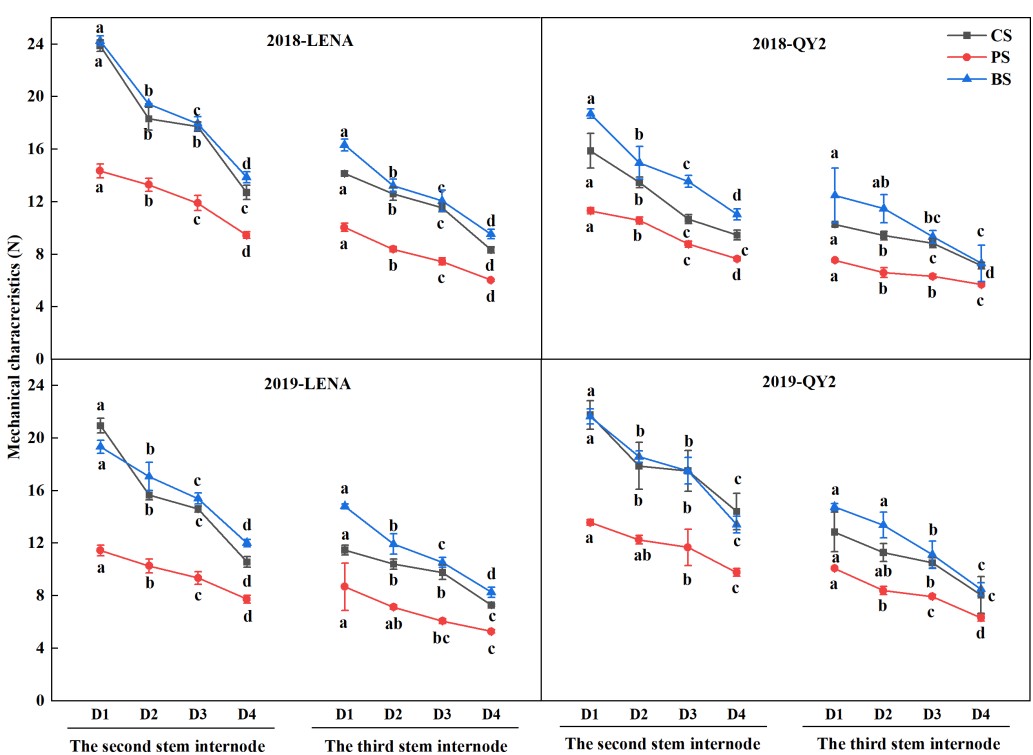

**Figure 1 Effects of planting density on mechanical characteristics of the two oat varieties grown in 2018 and 2019.** Different lowercase letters represent significant differences at $P < 0.05$.

**Table 2 Effects of planting density on the grass yield and its components of the two oat varieties grown in 2018 and 2019.**

| Varieties | D | 2018 | | | 2019 | | |
|---|---|---|---|---|---|---|---|
| | | SPW | FGY | HY | SPW | FGY | HY |
| LENA | D1 | 12.56a | 40.87b | 7.84b | 15.07a | 39.40d | 10.82d |
| | D2 | 11.13b | 45.09ab | 8.61ab | 14.54a | 46.31b | 13.62b |
| | D3 | 9.42c | 48.00ab | 8.87ab | 12.13b | 49.88a | 14.33a |
| | D4 | 5.92d | 49.69a | 9.69a | 7.54c | 41.64c | 12.07c |
| QY2 | D1 | 10.82a | 37.35b | 6.51b | 10.98a | 38.36b | 10.77b |
| | D2 | 9.17b | 46.75a | 8.68a | 9.08b | 48.64a | 11.53ab |
| | D3 | 7.53bc | 50.68a | 9.81a | 8.16bc | 50.34a | 12.59a |
| | D4 | 7.84c | 47.44a | 9.32a | 7.46c | 39.94b | 11.82ab |

**Notes.**

Different letters represent significant differences at $P < 0.05$.

SPW, single-plant weight (g); FGY, fresh grass yield (t/ha); HY, hay yield (t/ha).

### Grain yield and its components

Analyses of the grain yield and its components of the two varieties suggested that the lodging-resistant variety LENA exhibited reduced main panicle length, but increased grain yield and 1,000-grain weight, compared to the lodging-sensitive variety QY2 in two years (Table 3). With increasing planting density, the main panicle length, number of grains

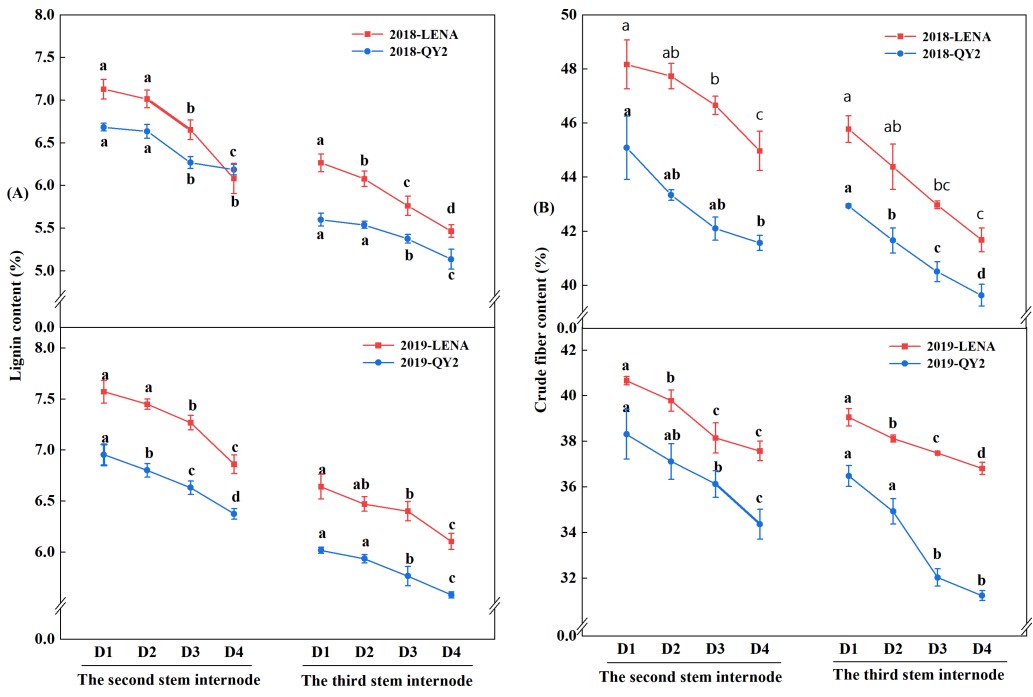

**Figure 2   Effects of planting density on lignin content (A) and crude fiber content (B) of the two varieties grown in 2018 and 2019.** Different lowercase letters represent significant differences at $P < 0.05$.

**Table 3   Effects of planting density on the grain yield and its components of the two varieties grown in 2018 and 2019.**

| Varieties | D | 2018 | | | | | 2019 | | | | |
|---|---|---|---|---|---|---|---|---|---|---|---|
| | | MPL | NGP | WGP | TGW | GY | MPL | NGP | WGP | TGW | GY |
| LENA | D1 | 16.13a | 112.67a | 3.23a | 31.95a | 5.96b | 16.73a | 109.00a | 3.19a | 32.26a | 6.03b |
| | D2 | 15.16ab | 81.58b | 2.78b | 31.70a | 6.63a | 15.85ab | 94.33b | 2.93a | 31.67ab | 6.98a |
| | D3 | 15.37ab | 54.92c | 1.96c | 30.92ab | 5.93b | 15.01b | 66.00c | 2.22b | 31.06b | 6.31ab |
| | D4 | 12.72b | 36.42d | 1.10d | 29.83b | 5.81b | 13.43c | 51.00d | 1.69c | 30.13c | 6.18b |
| QY2 | D1 | 17.81a | 88.58a | 2.86a | 28.15a | 4.05b | 18.25a | 94.67a | 3.51a | 28.23a | 4.13c |
| | D2 | 16.84ab | 82.26a | 2.19ab | 25.46a | 4.67a | 16.52b | 85.29b | 2.36b | 25.87b | 4.86a |
| | D3 | 16.45ab | 52.67b | 1.85ab | 20.14b | 4.25b | 15.92b | 58.04c | 1.55c | 21.89c | 4.51b |
| | D4 | 13.98b | 37.72c | 1.20b | 16.65c | 3.70c | 15.00b | 45.89d | 1.22c | 19.70d | 3.85d |

**Notes.**
Different lowercase letters represent significant differences at $P < 0.05$.
MPL, main panicle length (cm); NGP, number of grains per plant; WGP, weight of grains per plant (g); TGW, 1,000-grain weight (g); GY, grain yield (t/ha).

per plant, weight of grains per plant, and 1,000-grain weight of both varieties tended to decrease. Furthermore, grain yield tended to increase and then decrease, reaching maxima in D2 and minima in D4, and grain yield in D2 and D4 had significant difference ($P < 0.05$).

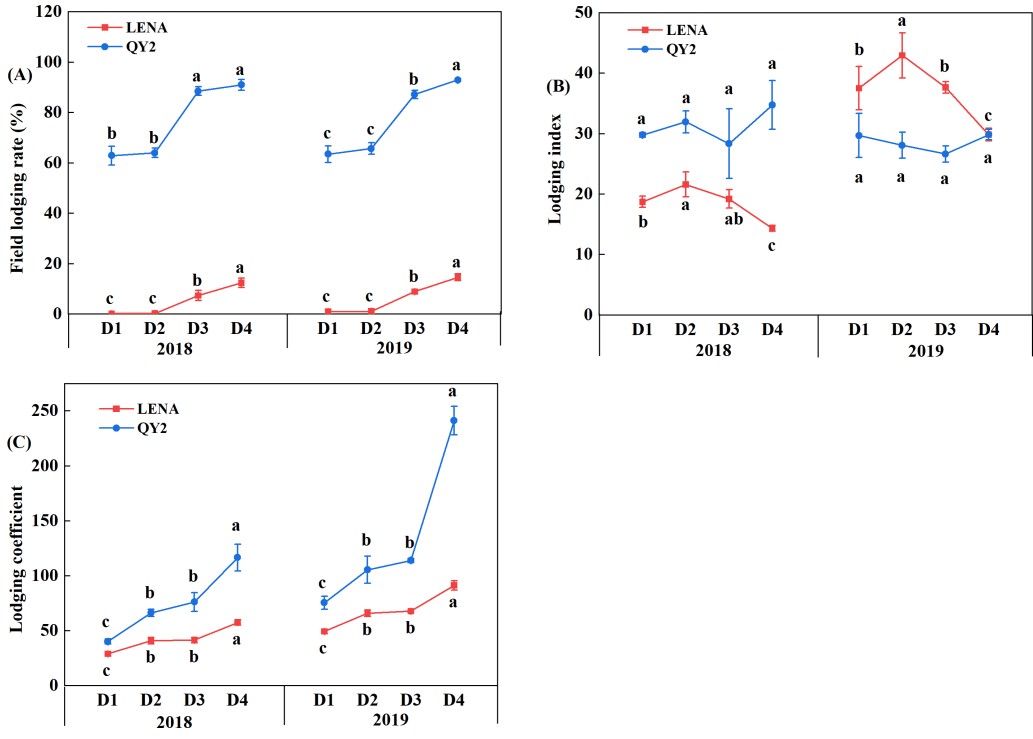

**Figure 3  Effects of planting density on the field lodging rate (A), lodging index (B), and lodging coefficient (C) of the two oat varieties grown in 2018 and 2019.** Different lowercase letters represent significant differences at $P < 0.05$.

## Analysis of variance

The effects of the growth year, oat variety, planting density, and their interactions on agronomic traits, mechanical characteristics, biochemical compositions, lodging, and yield and its components of oat are shown in Table 4. The results revealed that the growth year exhibited no significant effect on puncture strength, breaking strength, fresh grass yield, main panicle length and weight of grains per plant. Whereas the oat variety did not significantly impact puncture strength and fresh grass yield. The planting density exerted a significant effect on all parameters ($P < 0.05$). Moreover, the interaction between the growth year and oat variety significantly affected puncture strength, stem diameter, stem wall thickness and so on ($P < 0.01$). Similarly, the interaction between the growth year and planting density had a significant impact on root fresh weight, lignin, lodging coefficient, and fresh grass yield ($P < 0.01$). Furthermore, the interaction between oat variety and planting density significantly influenced stem diameter, stem wall thickness, and length of the stem internode ($P < 0.01$). Notably, the interaction among growth year, oat variety, and planting density was found to significantly affect root fresh weight, puncture strength, lodging coefficient, and hay yield ($P < 0.01$).

Liu et al. (2024), *PeerJ*, DOI 10.7717/peerj.17310

**Table 4  Analysis of variance.**

| Sources of variation | Mean squares of the measured traits | | | | | | | | | | |
|---|---|---|---|---|---|---|---|---|---|---|---|
| | PH | SD | WT | SL | HCG | RFW | CS | PS | BS | CF | LN |
| Y | $0.26^{**}$ | $0.19^{*}$ | $0.20^{**}$ | $65.03^{**}$ | $0.10^{**}$ | $0.57^{**}$ | $5.27^{**}$ | $0.02^{ns}$ | $0.37^{ns}$ | $577.55^{**}$ | $2.28^{**}$ |
| V | $0.44^{**}$ | $0.75^{**}$ | $1.13^{**}$ | $855.67^{**}$ | $0.16^{**}$ | $2.97^{**}$ | $20.09^{**}$ | $0.30^{ns}$ | $15.84^{**}$ | $129.62^{**}$ | $2.80^{**}$ |
| D | $0.03^{**}$ | $3.38^{**}$ | $0.06^{**}$ | $8.25^{**}$ | $0.003^{**}$ | $2.12^{**}$ | $90.52^{**}$ | $28.18^{**}$ | $111.46^{**}$ | $29.51^{**}$ | $0.94^{**}$ |
| Y*V | $0.05^{**}$ | $0.68^{**}$ | $0.03^{**}$ | $85.22^{**}$ | $0.01^{**}$ | $1.55^{**}$ | $106.78^{**}$ | $43.90^{**}$ | $65.82^{**}$ | $0.10^{ns}$ | $0.13^{**}$ |
| Y*D | $0.001^{ns}$ | $0.006^{ns}$ | $0.000^{ns}$ | $2.40^{*}$ | $0.07^{ns}$ | $0.04^{**}$ | $0.15^{ns}$ | $0.08^{ns}$ | $0.37^{ns}$ | $0.06^{ns}$ | $0.02^{**}$ |
| V*D | $0.002^{ns}$ | $0.42^{**}$ | $0.009^{**}$ | $21.19^{**}$ | $0.000^{ns}$ | $0.12^{**}$ | $3.12^{**}$ | $0.32^{*}$ | $0.60^{ns}$ | $0.52^{ns}$ | $0.04^{**}$ |
| Y*V*D | $0.000^{ns}$ | $0.01^{ns}$ | $0.001^{ns}$ | $2.47^{*}$ | $0.000^{ns}$ | $0.06^{**}$ | $0.96^{ns}$ | $0.55^{**}$ | $0.80^{ns}$ | $0.83^{ns}$ | $0.02^{*}$ |

| Sources of variation | FLR | LI | LC | SPW | FGY | HY | MPL | NGP | WGP | TGW | GY |
|---|---|---|---|---|---|---|---|---|---|---|---|
| Y | $12.65^{*}$ | $756.60^{**}$ | $540.85^{**}$ | $20.99^{**}$ | $24.21^{ns}$ | $149.37^{**}$ | $0.96^{ns}$ | $617.96^{**}$ | $0.42^{ns}$ | $6.74^{*}$ | $0.64^{**}$ |
| V | $60.43^{**}$ | $55.62^{**}$ | $709.27^{**}$ | $56.00^{**}$ | $0.36^{ns}$ | $4.36^{**}$ | $20.17^{**}$ | $693.42^{**}$ | $1.05^{*}$ | $754.57^{**}$ | $46.86^{**}$ |
| D | $384.76^{**}$ | $35.26^{**}$ | $324.96^{**}$ | $59.57^{**}$ | $245.40^{**}$ | $12.59^{**}$ | $24.68^{**}$ | $354.84^{**}$ | $8.07^{**}$ | $88.23^{**}$ | $1.85^{**}$ |
| Y*V | $1.22^{ns}$ | $195.35^{**}$ | $80.25^{**}$ | $18.53^{**}$ | $0.42^{ns}$ | $2.25^{ns}$ | $0.20^{ns}$ | $27.36^{ns}$ | $0.03^{ns}$ | $3.94^{ns}$ | $0.04^{ns}$ |
| Y*D | $2.12^{ns}$ | $9.76^{ns}$ | $43.90^{**}$ | $0.73^{ns}$ | $55.50^{**}$ | $1.71^{*}$ | $0.94^{ns}$ | $54.85^{ns}$ | $0.07^{ns}$ | $1.52^{ns}$ | $0.03^{ns}$ |
| V*D | $248.41^{**}$ | $93.95^{**}$ | $107.12^{**}$ | $13.02^{**}$ | $15.43^{ns}$ | $0.30^{ns}$ | $0.21^{ns}$ | $185.14^{**}$ | $0.18^{ns}$ | $38.99^{**}$ | $0.13^{ns}$ |
| Y*V*D | $1.92^{ns}$ | $5.82^{ns}$ | $27.55^{**}$ | $0.36^{ns}$ | $2.81^{ns}$ | $2.65^{**}$ | $0.23^{ns}$ | $56.71^{ns}$ | $0.27^{ns}$ | $1.30^{ns}$ | $0.007^{ns}$ |

**Notes.**

$^{*, **}$Denote significant differences at $P < 0.05$ and $P < 0.01$, respectively.

ns, represents not significant; Y, growth year; V, oat variety; D, planting density.

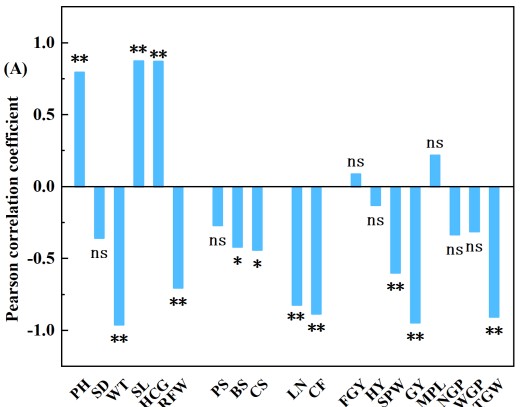
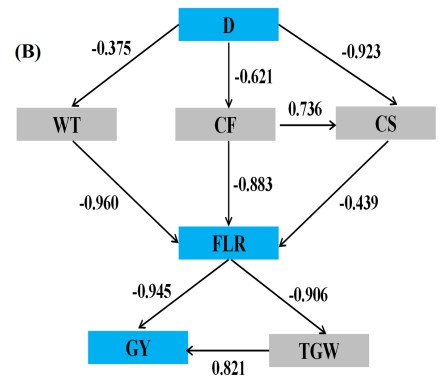

**Figure 4** Correlation analysis between field lodging rate and agronomic traits, stem mechanical characteristics, physiological indicators, and yield and its components (A); regression analysis of the planting density, lodging, yield, (B). * and ** denote significant differences at $P < 0.05$ and $P < 0.01$, respectively; ns represents no significant difference. The numerals in (B) represent normalization coefficients.

## Relationship among planting density, field lodging rate, and yield

Analyses of correlations between field lodging rate and agronomic traits, stem mechanical characteristics, biochemical compositions, and yield and its components revealed a highly significant, positive correlation of field lodging rate with plant height, length of the stem internode, and height of center of gravity; a highly significant, negative correlation of field lodging rate with stem wall thickness, root fresh weight, lignin content, crude fiber content, single-plant weight, grain yield, and 1,000-grain weight ($P < 0.01$); and a significant negative correlation of field lodging rate with compressive strength and breaking strength of stems ($P < 0.05$) (Fig. 4A). Further regression analysis was performed to select one indicator from each of the agronomic traits, stem mechanical characteristics, biochemical compositions, and yield and its components that exhibited the strongest correlation with lodging. The results indicated that planting density affected lodging by influencing stem wall thickness, crude fiber content, and compressive strength, whereas lodging altered grain yield by affecting 1,000-grain weight (Fig. 4B).

## DISCUSSION

Lodging involves root displacement and stem breaking. Both lodging index and lodging coefficient serve as comprehensive evaluation indices of lodging (*Wang et al., 2023*). In this study, we found that lodging index did not effectively indicate lodging characteristics of the oat varieties at different planting densities, because planting density affects root development, and lodging index ignores the influence of the root. In such cases, lodging coefficient can reflect lodging characteristics more comprehensively. We found that lodging coefficient increases with increasing planting density, indicated lodging resistance decreased, which is consistent with field lodging rate. Lodging resistance varied greatly between varieties, with some varieties remaining upright only at low densities, and lodging rate increasing with increasing density (*Gao et al., 2023*). Our study also yielded
similar findings, unlike previous studies, regardless of high or low planting density, QY2 consistently exhibited a higher lodging rate, whereas LENA showed the opposite trend. This suggests that under natural conditions, lodging is largely determined by varietal characteristics. In addition, we found no significant difference between the lodging rate at planting densities D1 and D2, indicating that increasing planting density within a certain range did not increase the risk of lodging.

Plant height, height of center of gravity and the length, diameter, wall thickness, and plumpness of the basal internodes are key morphological indicators of the strength of lodging resistance (*Argenta et al., 2021*). Therefore, in this study, we focused on investigating morphological characteristics of the above-ground second and third stem internodes, as well as plant height and height of center of gravity. We noted that planting density exhibited a significant effect on these traits ($P < 0.05$), and these traits were closely related to lodging rate. Specifically, the plant height, internode length and height of center of gravity were found to be significantly positively correlated with lodging rate, whereas stem wall thickness and root fresh weight were significantly negatively correlated with lodging rate ($P < 0.01$). These factors can serve as important indicators for evaluating oat stem lodging resistance.

Plant height of the two oat varieties increased from D1 to D2, probably because the reasonable planting density increased beneficial interactions between populations, thus optimizing the growth performance of the varieties. However, plant height decreased as planting density was further increased, because extremely high planting density altered land-resource use, thus limiting the growth of oat. Height of center of gravity, which exhibits a significant effect on lodging resistance, has been shown lower in lodging-resistant varieties compared to lodging-sensitive varieties (*Luo et al., 2022*). Indeed, height of center of gravity was lower in LENA compared to QY2, and it varied across planting densities, showing a trend consistent with plant height. The diameter and wall thickness of the second and third stem internodes decrease with increasing planting density, yet the correlation between stem diameter and lodging rate is not significant. This suggests that planting density primarily influences lodging by regulating stem wall thickness, with denser planting resulting in thinner wall thickness and higher lodging rate. Regression analysis further confirmed this observation. Additionally, root characteristics are also an important factor affecting lodging. High root biomass can enhance plant anchoring ability and reduce lodging occurrence. In this study, we found that increasing planting density reduced root fresh weight and increased the lodging rate.

Stem mechanical strength and stiffness—key factors affecting lodging—can be measured through puncture strength, breaking strength, and compressive strength of stem. The performance of these mechanical characteristics is determined by filler substances, such as lignin, cellulose, and hemicellulose in the stem (*Ookawa et al., 2010*; *Sun et al., 2022*). Lignin, cellulose, and hemicellulose, the main components of crude fiber, play an important role in crop lodging. Lignin content in stems can be used as an effective index for assessing the lodging resistance of intercropped soybeans, which was significantly positively correlated with the stem breaking strength and significantly negatively correlated with the actual lodging rate (*Liu et al., 2019*). Cellulose accumulation increased breaking

strength and lodging resistance of soybean basal stem, which were significantly negatively correlated with lodging rate (*Liu et al., 2016*). However, it has also been suggested that lodging is not related to the contents of lignin and cellulose in wheat stems, but rather to their arrangement in the stem cell wall and their interactions with each other (*Knapp, Harms & Volenec, 1987*).

Studies have shown that planting densities can regulate the synthesis of lignin and cellulose, which in turn affects the lodging (*Li et al., 2021*; *Zheng et al., 2017*). Appropriate low planting density can increase lignin-related enzyme activity and carbohydrate accumulation in stems, ultimately enhancing the lodging resistance of intercropped soybean (*Cheng et al., 2020*). In this study, planting density was found to have a highly significant effect ($P < 0.01$) on lignin and crude fiber contents, puncture strength, breaking strength, and compressive strength, with all of them decreased with increasing painting density, and except for puncture strength, all indicators were significantly negatively correlated with lodging rate ($P < 0.05$). Moreover, regression analysis indicated that the number of plants per unit area affected the accumulation of lignin and crude fiber and that high planting density decreased lignin and crude fiber contents due to insufficient growing space and nutrients, which in turn decreased the mechanical strength of the stem and ultimately increased the risk of lodging.

The production potential of a crop can be maximized by optimizing its population density (*Williams et al., 2021*). Reports have shown that increasing planting density can increase maize yield to some extent, mainly by taking full advantage of the population owing to an increased number of panicles per unit area. However, extremely high planting density will reduce the grain number and grain weight of the panicle (*Yang et al., 2021*). The grain yield of oilseed rape increases with increasing planting density, peaking at high density (*Khan et al., 2017*), and it has also been reported that without an increase in yield per unit area after reaching the saturation threshold due to intense intraspecific competition for resources (*Zhao et al., 2020*).

Oat is a special crop, with both high grass yield and high grain yield adding to its economic value. Hence, this study was proposed to examine the mechanism underlying the coordination of the nutritional and reproductive growth of oat by adjusting population density. We found that single-plant weight, main panicle length, number of grains per plant, weight of grains per plant, and 1,000-grain weight decreased with increasing planting density, but both grass yield and grain yield showed an increasing and then a decreasing trend, indicating that increasing plants per unit area could compensate for the loss caused by the decrease in yield per plant within a certain range. In addition to the direct effects of planting density, lodging can affect yield, especially grain yield by a highly significant margin ($P < 0.01$). The analysis in this study revealed that lodging mainly reduced grain yield by decreasing 1,000-grain weight, but the effect of lodging on grass yield was not significant. At planting density D3, the fresh grass yield and hay yield of both varieties reached the maxima, but the differences for planting density D2 were not significant, and both varieties exhibited the highest grain yield at planting density D2. Therefore, we suggest using D2 as a reasonable planting density for oat cultivation.

## CONCLUSIONS

The planting density significantly affected the growth, lodging, and yield of oat ($P < 0.05$). Planting density affects lodging by regulating morphological characteristics, such as plant height, height of center of gravity, stem wall thickness, internode length, and root fresh weight of oat. Additionally, it can impact stem mechanical strength by modulating the synthesis of lignin and crude fiber, thereby affecting lodging. Plant height, height of center of gravity, stem wall thickness, internode length, root fresh weight, breaking strength, compressive strength, lignin and crude fiber content, single-plant weight, grain yield and 1,000-grain weight can serve as important indicators for evaluating oat stem lodging resistance. Lodging mainly affects seed yield by influencing 1,000-grain weight but exhibited no significant effect on grass yield. In this study, we found that the planting density of $4.5 \times 10^6$ plants/ha (D2) was the best for improving lodging resistance, grass yield, and grain yield of oat.

### Funding

This work was supported by the Grass Seed Innovation and Its Role in Grassland Agricultural Systems (2023-NK-147). The funders had no role in study design, data collection and analysis, decision to publish, or preparation of the manuscript.

### Grant Disclosures

The following grant information was disclosed by the authors:
Grass Seed Innovation and Its Role in Grassland Agricultural Systems: 2023-NK-147.

### Competing Interests

The authors declare there are no competing interests.

### Author Contributions

- Lingling Liu conceived and designed the experiments, performed the experiments, analyzed the data, prepared figures and/or tables, and approved the final draft.
- Guoling Liang conceived and designed the experiments, performed the experiments, prepared figures and/or tables, and approved the final draft.
- Wenhui Liu conceived and designed the experiments, authored or reviewed drafts of the article, and approved the final draft.
- Zeliang Ju analyzed the data, authored or reviewed drafts of the article, and approved the final draft.

### Data Availability

The raw measurements are available in the Supplementary File.

## Supplemental Information

Supplemental information for this article can be found online at http://dx.doi.org/10.7717/peerj.17310#supplemental-information.

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
