# Peer review of "Variation and interrelationships in the growth, yield, and lodging of oat under different planting densities"

_PeerJ, doi:10.7717/peerj.17310_

## Round 0.1 · original submission · Major Revisions

Increasing lodging tolerance in oats is an important breeding goal, and your manuscript is the product of a useful study in this aspect. However, it is already a known fact that as the density of the plant increases, the rate of thinning of the stem and lodging will increase. You presented rational measurements to determine the lodging rate. It also makes sense to measure them at different plant densities. Mention a little more about the importance of the features you measured. The discussion section needs a lot of improvement. In the conclusion section, you mentioned the most appropriate plant density. Which genotype is this plant density suitable for? The one that can withstand lodging? Sensitive one to lodging? Additionally, you should also highlight which of your measurements may be the best selection criterion to determine lodging tolerance in oats.

·

Basic reporting

The article has been prepared according to the specified standards.

Experimental design

The trial design was prepared to cover the basic requirements in the field.

Validity of the findings

The findings were determined and explained accurately and appropriately according to scientific techniques.

Additional comments

It is appropriate to publish it after making the changes specified in the attachment.

·

Basic reporting

Clear, professional English is used throughout.
The article complies with professional standards of courtesy and expression.

The results and discussion are supported by the literature.

The materials and methods used in the research are appropriate.

Experimental design

The examined features support the result of the study and are sufficient.

Statistical analysis, evaluation of data, and presentation of tables are understandable and descriptive.

Validity of the findings

The suggestions were found to be guiding and useful for oat cultivation in the region.

Additional comments

The article meets the publication criteria.

·

Basic reporting

Lacks consistency. For instance in the background part of the abstract in line 17, the author used plural forms like “Oats are…” but coming to the introduction in line 47 they used singular forms like “Oat (Avena sativa L.) is…” which might confuse the reader. Please consider revising.
 Ambiguous sentences such as from line 48 - 50. I suggest to paraphrase as follows: The grain is rich in nutrients making it a preferable component of a healthy diet (Muhammad et al …). Please consider your long and ambiguous sentences short and clear as much as you can throughout the manuscript. You can also read your sentence from line 84 – 86 which is very long and ambiguous. What do you mean by technical measures in this statement? Why planting density here, since you told the reader from the previous sentence (Line 82 – 83) that the research gaps are oat varieties and high-quality nutrient resources. This means you should either work on new oat variety development or high-quality nutrients for your target production constraint, in my understanding lodging.
 Unfinished sentences or inappropriate use of although in line 80.
 Out of the context. For instance you mentioned the importance of oat then without showing the productivity of oat in the world and your region/country you simply introduce the production constraints. In line 86 the author also introduce environmental management which is a big issue and livestock farming development which is another area that the author goes out of the context. Please try to justify why you did your research on oat planting density and lodging very clearly based on the literatures.
 Unclear objectives. I suggest you the following three clear objectives for your study:
1. To study the effect of planting density on oat lodging
2. To study the effect of planting density on grain yield and yield related traits
3. To determine the optimum planting density for oat production
 What do you mean by theoretical basis in line 92? Although I am not sure what your institution or country demanded to boost oat production, It is not the theoretical rather the practical basis is important to increase any crop production.
 In the results section, particularly in line 224, the author introduced the concept of stiffness which they did not show in the method section. What stiffness indicates should be seen in your study methods?

Experimental design

As if you have done sets of experiments, you stated in line 96 “the experiments were” which is plural form. Whereas you did a single experiment over two years. So better to use singular form “the experiment was conducted”
 In line 96, please include the country name where the city and province is found.
 Inappropriate use of comma, For instance in line 98 the author use comma just before the word “period”. It confuses a lot I do not understand what to mean. Please consider paraphrasing the sentence.
 In line 124, it is indicated that root fresh weight was taken, but they did not say something about how they took. It will be good to clearly indicate the procedure of your root measurement.
 How did you measure the wall thickness? Please qualify it in line 131.
 The method is generally lacks clarity, for instance in line 156 and 158, the author tell us about digestion and heating for a certain minutes, but did not mention about the temperature they used for digestion and heating which is very important information.
 I think the authors are not clear about which mechanical strength measurement is best characterize the oat stalk (stem). What is the difference between the three strength measures you considered (Puncture, Breaking, and Compressive)? Are these all measured by the same instrument with the same sample?
 Move your method related to lodging from line189 – 195 to the end of “mechanical characteristics” section that is just before line 149.
 I do not think that “physiological indicators” could best fit for crude fiber and lignin content. Better to consider it. I suggest “Biochemical composition”.
 Generally the method needs serious revision. The language should be improved and the method should be clear and show pertinent information and procedures with appropriate citations.

Validity of the findings

The study presented sufficient data and could deliver pertinent information. However, most of the result are based on simple analysis. For instance the first two paragraphs of the results section are simple trend analysis without statistical test. I suggest to do exhaustive ANOVA and every statement should be based on the results of the statistical test. I mean start with the ANOVA result (Table 4) and interpret it exhaustively for example what does the significant interaction between variety and planting density imply and the three way interactions as well. Carefully interpret the main effects as well. Then complement with the trend analysis.
 Your table 4 is not appropriate, please present the mean square/variance values and show the significant difference indicates (strikes) as superscript.

Additional comments

I appreciate the efforts of the author to produce the manuscript. However, the write up lacks coherence and readability. Thus please consult your senior colleagues and read similar articles published and amend your manuscript based on the above comments I gave and beyond. The discussion is very shallow. Please compare your result with similar works within oat and/or other crops and show the novelty of your results.

---

## Round 0.2 · Minor Revisions

Resubmit your manuscript after making any minor corrections indicated by the reviewer.

·

Basic reporting

Clear and unambiguous English used throughout the text.
Literature references, field background, article structure, figures and tables are sufficient.

Experimental design

Experimental design and method described with sufficient detail and information.

Validity of the findings

The findings and discussion were explained in accordance with the essence of the research and the results were explained by discussion.

Additional comments

The requested changes have been made in sufficient quantities

·

Basic reporting

It would be good to change the title as follows “Variation and interrelationships in the growth, yield, and lodging of oat under different planting densities”

 Would you mind to amend the background part (Line 17 – 20) of your abstract as follows “Oat is a dual-purpose cereal used for grain and forage. The demand of oat has been increasing as the understanding of the nutritional, ecological, and economic values of oat increased. However, the frequent occurrence of lodging during the growing period severely affect the high yielding potential and the quality of the grain and forage of oat.”

 In the abstract section (line 38) you used etc., I do not think this is appropriate. Please mention all the remaining traits and avoid using etc. throughout your manuscript if any other

 In line 56 change the phrase “mechanized processing” to “mechanized harvesting”

 In line 385 change the word “effects” to “affected”

 In line 390 avoid etc. and mention the remaining traits

 In table 1 separate the first column into two with Varieties and Years as a column heading.

 In table 2 and 3, change the world “Items” to “Varieties”

 In table 4, I suggested to put the mean squares (variances) that means the sum squares of each sources of variation divided by the respective degree of freedom and put the asterisk to indicate whether it is statistically significant, not the F-values please. Then change the word “Items” to Sources of variation”, “F value” to “Mean squares of the measured traits”

 In table 4 again in the last row (Y*V*---), I think something missed here in the interaction

Experimental design

well organized

Validity of the findings

Valid

---

## Round 0.3 · accepted · Accept

You have followed the reviewers' and editorial recommendations, and my decision for the manuscript is its acceptance.